# Rap1 Is Involved in Angiopoietin-1-Induced Cell-Cell Junction Stabilization and Endothelial Cell Sprouting

**DOI:** 10.3390/cells9010155

**Published:** 2020-01-08

**Authors:** Vanda Gaonac’h-Lovejoy, Cécile Boscher, Chantal Delisle, Jean-Philippe Gratton

**Affiliations:** Department of Pharmacology and Physiology, Faculty of Medicine, Université de Montréal, Montreal, QC H3C 3J7, Canada

**Keywords:** angiopoietin-1, angiogenesis, Rap1, VE-cadherin, migration, adhesion, endothelial cells

## Abstract

Angiopoietin-1 (Ang-1) is an important proangiogenic factor also involved in the maintenance of endothelial-barrier integrity. The small GTPase Rap1 is involved in the regulation of adherens junctions through VE-cadherin-mediated adhesion, and in endothelial permeability. While many studies established that Rap1 activation is critical for endothelial cell–cell adhesions, its roles in the antipermeability effects of Ang-1 are ill-defined. Thus, we determined the contribution of Rap1 to Ang-1-stimulated angiogenic effects on endothelial cells (ECs). We found that Rap1 is activated following Ang-1 stimulation and is required for the antipermeability effects of Ang-1 on EC monolayers. Our results also revealed that Rap1 is necessary for EC sprouting stimulated by Ang-1 but had no significant effect on Ang-1-induced EC migration and adhesion. In contrast, downregulation of VE-cadherin markedly increased the adhesiveness of ECs to the substratum, which resulted in inhibition of Ang-1-stimulated migration. These results revealed that Rap1 is central to the effects of Ang-1 at intercellular junctions of ECs, whereas VE-cadherin is also involved in the adhesion of ECs to the extracellular matrix.

## 1. Introduction

Angiogenesis is the process by which new blood vessels emerge from existing ones. It is involved in many biological processes, including embryonic development, reproduction, and wound repair [1,2]. However, an imbalance in the angiogenic process participates in the establishment of many malignant diseases such as cancer, and inflammatory, ischaemic, and immune disorders. Angiopoeitin-1 (Ang-1) is an angiogenic growth factor that plays an essential role in endothelial cell (EC) survival, migration, sprouting, and tubule formation through EC-specific receptor Tie2 [3,4].

Dynamic regulation of the endothelial barrier is crucial for angiogenesis during development to promote the maturation of blood vessels. This barrier is preserved by adherens junctions (AJs) and tight junctions (TJs), both of which are linked to the actin cytoskeleton, and it maintains tissue integrity. Vascular endothelial-cadherin (VE-cadherin), a cell-adhesion molecule that forms Ca^2+^-dependent homodimer interactions across the cell membranes of neighboring cells, is at the heart of AJs [5]. VE-cadherin has a cytoplasmic tail that links the actin cytoskeleton through interactions with several cytoplasmic proteins including β-catenin and participates in transmitting signals that relate to the adhesive function of VE-cadherin and to the outside–in signaling in ECs, both pivotal for angiogenesis [5]. The small GTPase Rap1 is one of the essential signaling intermediates mediating the functions of VE-cadherin.

Rap1 activity is spatiotemporally regulated by guanine nucleotide exchange factors (GEFs) and GTPase activating protein (GAPs) and has been clearly described as an important modulator of cell–matrix and cell–cell adhesions in many cell types, including ECs [6,7]. Recent research has shown that, in ECs, GTP-bound Rap1 can regulate angiogenesis and the endothelial barrier through the fine-tuning of vascular permeability [8,9]. AJ assembly is enforced by Rap1 through the direct implementation of VE-cadherin adhesion and junction integrity via several Rap1 effectors, including KRIT1, RASIP1, and Afadin [10,11,12,13,14]. In ECs that have not yet formed stable junctions with neighboring cells, Rap1 plays an active role by fostering the endothelial barrier through the direct regulation of VE-cadherin adhesion and junction integrity via KRIT1 [14]. Active Rap1 can also indirectly control the endothelial barrier through the dynamic reorganization of the actomyosin cytoskeleton and suppression of the Rho-ROCK pathway [7]. AJ stabilization is triggered by a number of antipermeability factors such as Ang-1, which stabilizes blood vessels [15,16,17]. It is well established that Ang-1 promotes endothelial-barrier integrity by counteracting the propermeability effect of vascular endothelial growth factor (VEGF) on ECs [15,17,18]. We previously showed that Ang-1 acts, at least in part, through the inhibition of VEGF-stimulated nitric oxide release [15]. The role of Rap1 in the formation and reinforcement of AJs is well-established. However, its role as a downstream effector of Ang-1 on ECs has never been examined.

Herein, we determine the contribution of Rap1 to Ang-1/Tie2-stimulated angiogenic effects on ECs. We found that downregulation of Rap1 attenuated the antipermeability effects of Ang-1, and that Rap1 is necessary for Ang-1-stimulated EC sprouting. Surprisingly, downregulation of Rap1 had no significant effect on the migration of ECs, in contrast to the downregulation of VE-cadherin, which affected both processes. These results highlight the involvement of Rap1 in the effects of Ang-1 on EC junctions and in angiogenic sprouting.

## 2. Materials and Methods

### 2.1. Cell Culture

Bovine aortic endothelial cells (BAECs) obtained from VEC Technologies (Rensselaer, NY, USA) were cultured in Dulbecco modified Eagle medium (DMEM) supplemented with 10% fetal bovine serum (HyClone, GE Healthcare Life Sciences, Pistacaway, NJ, USA), 2.0 mM L-glutamine, 100 U/mL penicillin, and 100 μg/mL streptomycin.

### 2.2. Transfections and Cell Treatments

To silence Rap1 expression, a siRNA against both Rap1a and Rap1b was synthesized. SiRNA sequences targeting bovine Rap1 were 5′-GAAAGCAAGUUGAAGUAGAUU-3′ (siRap1_1; used throughout) and 5′-GCAAUGAGGGAUUUGUACAUU-3′ (siRap1_2). The siRNA sequence targeting bovine VE-cadherin was 5′-GCACAUUGAUGAAGAGAAAUU-3′. An unrelated scrambled siRNA was used as control, 5′-AUGAACGUGAAUUGCUCAAUU-3′ (Dharmacon, Chicago, IL, USA). For some experiments, BAECs were transfected with expression plasmids GFP-RAPL (provided by Dr. Michael Gold, University of British Columbia, Canada) and FAK-GFP (from Dr. Ivan R. Nabi, University of British Columbia, Canada) to visualize and quantify focal adhesions. SiRNAs and expression plasmids were transfected in BAECs using Lipofectamine 2000, according to manufacturer protocol (Invitrogen, Carlsbad, CA, USA). Then, 48 h after transfections, cells were starved overnight in serum-free media before stimulation by Ang-1. Ang-1 was obtained from R&D System (Minneapolis, MN, USA) and reconstituted at 100 ng/mL in sterile PBS with 0.1% bovine serum albumin (BSA).

### 2.3. Endothelial Permeability Assay

Permeability across endothelial cell monolayers was measured using type I collagen-coated Transwell units (6.5 mm diameter, 3.0 μm pore size polycarbonate filter; Corning Costar, Tewksbury, MA, USA). BAECs were plated 48 h after transfection at a density of 200,000 cells per well and cultured for 4 days until the formation of a tight monolayer. Cells were serum-starved for 1 h in DMEM containing 1% BSA. Ang-1 (100 ng/mL) was added to the upper chambers in the presence of 1mg/mL FITC-labeled dextran (40 kDa molecular mass; Invitrogen). Endothelial permeability was measured by collecting a 50 μL sample from the lower compartment, which was diluted with 300 μL phosphate-buffered saline (PBS) and measured for fluorescence at 520 nm when excited at 492 nm with a Wallac Victor ^3^V spectrophotometer (PerkinElmer, Boston, MA, USA).

### 2.4. Antibodies and Immunoblotting

For immunofluorescence and immunoblot analyses, we used Alexa-coupled antibodies from Invitrogen and horseradish peroxidase (HRP)-coupled antibodies from Jackson ImmunoResearch Laboratories (West Grove, PA, USA), respectively. Primary antibodies used were rabbit anti-Rap1 (recognizing both Rap1a and Rap1b; Cell Signaling, Danvers, MA, USA), goat anti-VE-cadherin and rabbit anti-Tie2 from Santa Cruz Biotechnology (Santa Cruz, CA, USA), mouse anti-phosphotyrosine (clone 4G10; Millipore Sigma, Burlington, MA, USA), and anti-β-actin (Cell Signaling) and mouse anti-eNOS (BD Biosciences, San Jose, CA, USA).

For immunoblotting, cells were solubilized with a lysis buffer containing 1% Nonidet P-40, 0.1% sodium dodecyl sulfate (SDS), 0.1% deoxycholic acid, 50 mM Tris-HCl (pH 7.4), 0.1 mM EGTA, 0.1 mM EDTA, 20 mM sodium fluoride, 1 mM sodium pyrophosphate tetrabasic, and 1 mM sodium orthovanadate. Lysate were incubated for 30 min at 4 °C, centrifuged at 14,000× *g* for 10 min, boiled in SDS sample buffer, separated by SDS-polyacrylamide gel electrophoresis, transferred onto a nitrocellulose membrane (Hybond-ECL; GE Life Sciences, Pittsburg, PA, USA), and Western-blotted. Antibody detection was performed with HRP-coupled antibodies from Jackson Laboratories and using the Image Quant LAS4000 chemiluminescence-based detection system (enhanced chemiluminescence; GE Life Sciences).

### 2.5. Immunofluorescence Microscopy

BAECs were cultured on 0.1% gelatin-coated coverslips (100,000 cells per coverslip) and transfected as previously described. Cells were serum-starved overnight and stimulated for 30 min with Ang-1. Cells were fixed for 20 min in serum-free DMEM containing 4% paraformaldehyde (PFA). Once fixed, cells were rinsed with PBS and permeabilized with 0.1% Triton for 5 min. Fixed cells were then incubated for 1 h with primary antibodies in 1% BSA in PBS, followed by 1 h incubation with the appropriate secondary antibodies labeled with Alexa Fluor 488 and/or 568. Coverslips were mounted on slides using Fluoromount (Sigma-Aldrich, St-Louis, MO, USA) and observed using a Zeiss LSM 800 confocal laser-scanning microscope. Images were assembled using Photoshop CS5 (Adobe Systems, San Jose, CA, USA). To quantify focal adhesions (FAs), BAECs were transfected with FAK-GFP and fixed after 48 h. Quantifications were performed using ImageJ version 1.49 (NIH, Bethesda, MD, USA) by applying a threshold on the GFP level and quantifying the number of GFP-positive FAs per cell. A total of 20 cells were quantified for each condition.

### 2.6. Rap1 Activation Assay and Immunoprecipitation

Rap1 activation was determined using an established pull-down method based on the binding of a GST fusion protein containing the Rap-binding domain of RalGDS (RalGDS-RBD/GST) to the active GTP-bound form of Rap1. TOPF10 *Escherichia coli* were transformed with expression vector pGEX-RalGDS-RBD, and RalGDS-RBD/GST fusion proteins (from Dr. Michael Gold, University of British Columbia, Canada) were induced with 0.1 mM isopropyl-1-thio-β-D-galactopyranoside (IPTG). Bacteria were then resuspended in a 50 mM Tris-HCl (pH 7.4) 50 mM NaCl, 1% Triton X-100, 1 mM protease inhibitor cocktail (Roche Life Sciences, Indianapolis, IN, USA) and 1% Nonidet P40, and sonicated. RalGDS-RBD/GST fusion proteins were purified from the sonicated supernatant by incubation with glutathione-coupled Sepharose 4B beads (Sigma-Aldrich) overnight at 4 °C. The beads were washed 3 times in a lysis buffer, and the amount of bound fusion proteins was estimated by SDS-PAGE and Coomassie Blue staining. BAECs were lysed in 1% Nonidet P40, 50 mM Tris-HCl, 0.1 mM EDTA, 0.1 mM EGTA, 0.1% SDS, 0.1% deoxycholic acid, 20 mM sodium fluoride, 1 mM sodium pyrophosphate tetrabasic, and 1 mM sodium orthovanadate. Aliquots of glutathione–Sepharose beads containing about 50 µg of RalGDS-RBD/GST proteins were then used to precipitate GTP-bound Rap1 from cell lysate supernatants by incubation for 1 h at 4 °C with gentle rotation. The beads were then washed 3 times with an excess of lysis buffer. The complexes were precipitated, boiled in SDS sample buffer, and bound Rap1 was revealed by immunoblotting.

For immunoprecipitation, cells were solubilized in a lysis buffer containing 1% Nonidet P-40, 0.1% sodium dodecyl sulfate (SDS), 0.1% deoxycholic acid, 50 mM Tris-HCl (pH 7.4), 0.1 mM EGTA, 0.1 mM EDTA, 20 mM sodium fluoride, 1 mM sodium pyrophosphate tetrabasic, and 1 mM sodium orthovanadate. Soluble proteins were incubated with anti-Tie2 antibodies (2 μg) at 4 °C overnight. Protein A-Sepharose (Sigma-Aldrich; 50 µL of a 50% slurry) was added and incubated for an additional hour. The immune complexes were precipitated and boiled in SDS sample buffer, and phosphotyrosine levels were revealed by anti-phosphotyrosine (4G10) immunoblotting.

### 2.7. Migration Assay and Time-Lapse Video Microscopy

Cells were first transfected with siRNAs, and then left for 48 h to recover and reach 90% confluency. BAECs were starved overnight in 12 well plates. Transfected cells were incubated with fluorescent vital Hoechst dye for 10 min before performing scratches with a 10 μL pipette tip on the confluent monolayer. Cell movements were recorded using an Axio-Observer Z1 microscope (Zeiss, Jena, Germany) equipped with an AxioCam MrM camera (Zeiss) and programmed to capture a frame every 10 min of the migration period (6 h). Temperature was maintained at 37 °C, and the atmosphere within the chamber was kept at 5% CO_2_/95% air throughout the experiment. Nuclei of cells at the leading edge were traced by time-lapse video microscopy with the Cell Tracker plugin of ImageJ (NIH, Bethesda, MD, USA), and nuclei tracks were analyzed with the Manage Cell Tracks plugin of Icy Software (Institut Pasteur, Paris, France). Total displacement (total distance migrated) and net displacement (distance from the cell’s starting point) were obtained from each of the tracks, and statistical analyses were performed as described below.

### 2.8. Spheroid-Based In Vitro Angiogenesis Assay

Then, 24 h after siRNA transfection, spheroids of BAEC were generated as previously described [16]. Briefly, transfected BAECs were cultured in DMEM containing 25% methocell, 10% FBS, 2.0 mM L-glutamine, 100 U/mL penicillin, and 100 μg/mL streptomycin in a U-shaped 96 well plate for 24 h to allow spheroid formation. Spheroids were transferred in a complete medium containing 45% collagen pH 7.4 and 20% Methocel (Sigma-Aldrich), and cultured for 24 h before fixation in 4% PFA. Images of the spheroids were taken after 24 h of sprouting using a camera mounted on an inverted Zeiss Axio observer A2 with a 10×/0.25 NA objective. The extent of capillary sprouting was quantified by measuring average sprout length and number of sprouts growing out of each spheroid using ImageJ 1.49v software. At least 10 spheroids per condition were analyzed, and 3 independent experiments were performed.

### 2.9. Cell-Adhesion Assay

BAEC adhesion was assayed in 96 well plates precoated with 1% gelatin overnight, as described, with minor modifications [19]. Then, 48 h after transfection, 50,000 cells were plated and left to adhere at 37 °C for 1 h. Adherent cells were stained with 0.2% crystal violet (Sigma-Aldrich) and 20% methanol in PBS, and adhesion was quantitated by measuring absorbance at 550 nm with a Wallac Victor ^3^V microplate reader (PerkinElmer). Each experiment point was performed in triplicate.

### 2.10. Statistical Analysis

Data are represented as means ± SEM. Two-tailed independent Student’s *t* tests were used when comparing 2 groups. Comparisons between multiple groups were made using one-way ANOVA followed by post hoc Bonferroni’s multiple-comparison test among groups using Prism 5 software (GraphPad, San Diego, CA, USA); *p*-value < 0.05 was considered statistically significant.

## 3. Results

### 3.1. Ang-1 EC Stimulation Activates Rap1

We investigated the effect of Ang-1 BAEC stimulation on Rap1 activation using a Rap1 activity assay. Ang-1 rapidly increased the binding of active GTP-bound Rap1 to the GST fusion protein containing the Rap-binding domain of RalGDS as determined by pull-down assay (Figure 1A). Quantifications revealed that Rap1 activation was tripled after 5 min of Ang-1 stimulation and sustained for at least 30 min (Figure 1B). Activation of Rap1 after Ang-1 stimulation was also determined by examining the cellular localization of GFP tagged RAPL, a downstream effector protein of Rap1. Following stimulation with Ang-1 of BAECs transfected with RAPL-GFP expression vectors, RAPL-GFP translocated to the plasma membrane in close proximity to VE-cadherin staining at cell junctions (Figure 1C). The colocalization coefficient of GFP and VE-cadherin immunofluorescence was increased twofold in cells treated with Ang-1 compared to untreated cells (Figure 1D). Collectively, these results indicated that Ang-1 BAEC stimulation leads to Rap1 activation, which promotes the relocalization of Rap1 effector RAPL at EC junctions.

### 3.2. Rap1 Required for Ang-1-Stimulated Stabilization of EC Junction

To determine whether Rap1 is essential for Ang-1-induced stabilization of the endothelial barrier, we examined the permeability of BAEC monolayers to macromolecules in cells with downregulated Rap1 expression. Transfection of BAECs with siRNA-targeting Rap1 (siRap1) decreased its protein expression by at least 75% when compared to BAECs transfected with control siRNA (siCT), as determined by immunoblot (Figure 2A and Appendix A). First, we determined if Rap1 downregulation affected the activation of Ang-1 receptor Tie2 following Ang-1 stimulation. BAECs transfected with siCT or siRap1 were stimulated with Ang-1 (100 ng/mL; 30 min), and Tie2 activation was monitored by Tie2 immunoprecipitation followed by anti-phosphotyrosine immunoblotting. Ang-1 BAEC stimulation caused Tie2 autophosphorylation on tyrosine residues in siCT-transfected cells, which was not affected in Rap1-downregulated cells (Figure 2B). These results indicated that Rap1 acts downstream of Tie2 to modulate the effects of Ang-1 in ECs.

Then, we examined the permeability of BAEC monolayers to fluorescein isothiocyanate (FITC)-labeled dextran (40 kDa) in cells transfected with siCT or siRap1. As expected, Ang-1 stimulation inhibited the transendothelial permeability of control-transfected BAECs (siCT) when compared to unstimulated cells (Figure 2C). Importantly, the reduction in transendothelial permeability induced by Ang-1 was inhibited in cells transfected with siRap1. To further document the involvement of Rap1 in the stabilization of endothelial junctions induced by Ang-1, we examined the reorganization of cell junctions after Ang-1 stimulation in Rap1-downregulated cells. By visualizing VE-cadherin localization in BAECs using immunofluorescence, we determined the linearity of intercellular junctions in ECs transfected with siCT or siRap1 (Figure 2D,E). The junctional linearity index was defined as the ratio of total junctional length over net linear junctional length at the periphery of cells. BAECs transfected with siCT had a VE-cadherin positive junction linearity index of around 0.7, whereas cells subjected to 30 min of Ang-1 had a linearity closer to 1, suggesting a tightening of intercellular junctions, which is consistent with a reduction in endothelial permeability. On the other hand, cells transfected with siRap1 had a significantly decreased linearity index when compared to untreated siCT-transfected cells, in agreement with Rap1 promoting cell–cell junction stability (Figure 2E). Furthermore, Ang-1-induced linearization of EC junctions was abolished in siRap1-transfected cells. Together, these observations suggested that Rap1 plays a significant role in the Ang-1-induced stabilization of EC junctions and the inhibition of transendothelial permeability.

### 3.3. Silencing of Rap1 or VE-Cadherin Inhibits EC Sprouting

To examine the involvement of Rap1 in Ang-1-stimulated angiogenesis in vitro, we used a three-dimensional spheroidal system of EC sprouting. EC spheroids were set in collagen gels, and the outgrowth of capillary-like structures was analyzed following Ang-1 treatment. In cells transfected with siRNA against Rap1, the Ang-1-induced sprouting of BAECs was significantly blocked (Figure 3B–D). Not only were fewer sprouts per spheroid in Rap1-silenced cells stimulated with Ang-1, sprout length was also significantly reduced (Figure 3C,D). These data show that Rap1 activity is necessary to promote Ang-1-induced sprouting of ECs in vitro.

Since Rap1 acts at VE-cadherin-based EC junctions, we examined if the downregulation of VE-cadherin had similar functional consequences on EC sprouting as those of Rap1 downregulation. For this purpose, we first transfected cells with a siRNA against VE-cadherin (siVE-cadherin) or siCT and confirmed at least 75% reduction in protein expression (Figure 3A). We then performed EC sprouting assays and quantified the endothelial sprouts from spheroids in response to Ang-1 transfected with siVE-cadherin. Similar to BAEC spheroids with silenced Rap1 expression, silencing VE-cadherin significantly blocked the Ang-1-induced sprouting of BAECs, reducing sprout length and number of sprouts per spheroids (Figure 3E–G).

### 3.4. Rap1 Not Necessary for Ang-1-Stimulated EC Migration

Next, we examined the role of Rap1 in Ang-1-induced EC migration. As we previously showed, Ang-1 induces an increase in total cell migration and net cell displacement of BAECs determined by live-cell tracking (Figure 4A–C) [16]. Interestingly, we did not observe any effect of siRap1 transfection on Ang-1-stimulated BAEC migration since cells behaved in an identical manner as cells transfected with siCT. This suggests that Rap1 is dispensable for Ang-1-stimulated EC migration (Figure 4A–C).

We then examined the effects of VE-cadherin downregulation on BAEC migration stimulated with Ang-1. In contrast to Rap1-downregulated cells, we observed a significant inhibition of Ang-1-stimulated cell migration in BAECs transfected with siVE-cadherin (Figure 4D,E). Total and net displacement (Figure 4D,E) of Ang-1-stimulated cells within the wounded area of the monolayers were markedly reduced in VE-cadherin-downregulated BAECs. Interestingly, basal migration was increased in siVE-cadherin-treated ECs. Overall, these data demonstrate that, in contrast to Rap1, VE-cadherin is required for Ang-1-induced EC migration.

### 3.5. Role of Rap1 and VE-Cadherin in Endothelial Cell Adhesion

To understand how Rap1 and VE-cadherin are differently involved in Ang-1 responses, we examined cellular adhesion to the extracellular matrix (ECM), which is an essential component of cell migration. Similar to its effect on EC migration, Rap1 downregulation had no effect on BAEC adhesion to the ECM in the presence of Ang-1 (Figure 5A). Interestingly, the basal adhesion of cells transfected with siVE-cadherin was increased compared to BAECs transfected with siCT. Moreover, downregulation of VE-cadherin prevented a further increase in adhesion stimulated by Ang-1 (Figure 5A). Since tyrosine phosphorylation of focal adhesion kinase (FAK) is linked to cell adhesion to the ECM and in FA remodeling, we examined the overall levels of tyrosine phosphorylation in BAECs, which is generally associated with FAK phosphorylation. We found that, in control-transfected cells, Ang-1 stimulation increased total phosphotyrosine staining (4G10 antibody), in particular at the cellular edges and in ruffle-like structures. Downregulation of Rap1 had no effect on Ang-1-induced EC phosphotyrosine staining. In contrast, siVE-cadherin-transfected cells had elevated basal phosphotyrosine staining, which was not further increased upon Ang-1 stimulation (Figure 5B). To directly monitor FA remodeling, we transfected BAECs with FAK-GFP, which was localized to FAs, and we assessed FA number per cell and their size. As we previously described, Ang-1 stimulation of control-transfected ECs increased the number of FAs and decreased their average size, which is consistent with increased cell adhesion (Figure 5C–E) [20]. Downregulation of Rap1 slightly increased basal and Ang-1-stimulated FA numbers in BAECs (Figure 5C,D) without affecting the Ang-1-stimulated reduction in FA size (Figure 5E). In siVE-cadherin-transfected cells, a marked increase in FA numbers and a decrease in the size of FAs was observed basally. These effects were enhanced in Ang-1-stimulated BAECs (Figure 5C–E). Overall, these results suggest that Rap1 is not involved in EC migration and adhesion stimulated by Ang-1, while VE-cadherin depletion results in a marked increase of adhesion, which prevents EC migration stimulated by Ang-1.

## 4. Discussion

Various angiogenic factors such as thrombin, sphingosine 1-phosphate, and VEGF rely on Rap1 activity to regulate endothelial permeability, migration, tube formation, integrin-mediated adhesion, and EC polarity [21,22,23,24,25,26,27]. The present study reveals the role of Rap1 in the proangiogenic effects of Ang-1 on ECs. Rap1 is activated in response to Ang-1 and is necessary for Ang-1-induced tightening of EC junctions and inhibition of endothelial permeability. Both Rap1 and VE-cadherin are required for EC sprouting in vitro, but only VE-cadherin is involved in the Ang-1 stimulation of cell adhesion and migration. Thus, these results revealed that Rap1 is central for the effects of Ang-1 at intercellular junctions of ECs, whereas VE-cadherin is also involved in EC adhesion to the extracellular matrix and, consequently, endothelial cell migration.

Ang-1 is well-known for its stabilizing effects on EC junctions and the inhibition of endothelial permeability [15,17,18]. Silencing Rap1 attenuated the ability of Ang-1 to reduce EC permeability and to stimulate EC sprouting, suggesting that Rap1 is responsible for coupling the Tie2 receptor to multiple downstream pathways in ECs. We therefore identified a novel molecular intermediate involved in Tie2 signaling where Ang-1 stimulation activates the small GTPase Rap1 in its GTP bound state, which consequently induces the activation of signaling pathways involved in the formation of cell–cell junctions between ECs. These results are in line with a previous study showing that overexpression of Rap1GAP1, a GTPase-activating protein that inhibits Rap1 activity, blocked the angiogenic sprouting and tube-forming activity of endothelial cells [22]. Moreover, others showed that, in mice, deletion of a single allele of *Rap1a* or *Rap1b* results in defective angiogenesis [22,28]. Our study solely relied on in vitro angiogenic assays to demonstrate the importance of Rap1 in Ang-1-mediated effects. These results should be confirmed using *Rap1a-* and *Rap1b*-deficient mice to examine their roles in Ang-1-driven angiogenesis in vivo.

Interestingly, the role of Rap1 in the dissociation of EC junctions and induction of endothelial permeability induced by VEGF was previously reported [29,30]. Herein, we report that Rap1 also mediates the tightening of cell–cell junctions and the inhibition of endothelial permeability induced by Ang-1, which is similar to its roles in the enhancement of the endothelial-barrier function by cyclic AMP mobilizing agonists [9,23]. This indicates that Rap1 acts as a central regulator of the barrier properties of the endothelium. However, we also demonstrated that Rap1 is dispensable for Ang-1-stimulated EC migration and adhesion to the substratum. This is in contrast with previous studies showing that Rap1 activation is involved in basic fibroblast growth factors (bFGF) and VEGF-stimulated migration of ECs [22,28]. Ang-1 and Tie2 were shown to directly interact with integrins or with the extracellular matrix, supporting that Rap1-independent mechanisms could be involved in the enhancement of cell–matrix interactions by Ang-1 [31,32,33]. This suggests that distinct signalling mechanisms are activated by Tie2 in ECs to modulate cell–matrix and cell–cell contacts, and that the function of Rap1 in Ang-1-stimulated angiogenesis is mostly related to its role in cell–cell adhesions of ECs.

In contrast to the effects of Rap1 silencing on EC migration, we found that ECs with silenced VE-cadherin showed a marked increase in basal migration in the absence of Ang-1 stimulation (Figure 4D,E). This result corroborates studies showing that cells expressing VE-cadherin migrate less than VE-cadherin-negative cells [34]. Interestingly, Ang-1 stimulation of VE-cadherin-silenced ECs did not further stimulate their migration. We also found that depleting VE-cadherin from BAECs caused a significant increase in cell adhesion and markedly enhanced the remodeling of focal adhesions. This effect of VE-cadherin depletion on EC adhesion could explain the inhibition of cell migration stimulated by Ang-1. Indeed, increased cell adhesion to the matrix above a certain threshold may reduce cell migration. VE-cadherin was previously linked to the control of cell spreading and focal adhesions by regulating RhoA activity [35]. RhoA activity is likely increased in VE-cadherin-depleted ECs, which contributes to the increase in cell adhesion to the ECM. On the other hand, Rap1 activity is not necessary for Ang-1-induced adhesion of ECs; therefore, cell migration is not affected by Rap1 knockdown. It is clear that both Rap1 and VE-cadherin are essential for regulating the dynamics of adherens junction assembly and endothelial permeability, and that Ang-1 stimulation induces the cooperation between these two proteins during angiogenic sprouting. The fact that Rap1 and VE-cadherin suppression results in the inhibition of EC sprouting from spheroids suggests that Rap1 and VE-cadherin cooperate during angiogenic sprouting. Indeed, Rap1 activation was shown to recruit the Raf-1/Rok-alpha complex at nascent VE-cadherin cell–cell junctions to strengthen EC cohesion during sprouting [36].

Similar to other GTPases, Rap1 activity is spatiotemporally regulated by guanine exchange factors (GEFs) and GTPase activating proteins (GAPs) [37,38]. Rap1 activity rapidly increases following Ang-1 EC stimulation. However, the underlying mechanism is uncertain. It is likely that a GEF, which acts downstream of the Ang-1 receptor, acts as an intermediate. Furthermore, the roles of C3G and PDZ-GEF (RA-GEF1), two known Rap1 GEFs, were studied in endothelial cells and in vascular development in mice [39,40,41]. Genetic deletion of both of these Rap1 GEFs in mice results in defective vascular morphogenesis [40,41]. Further studies are required to fully determine the mechanism of Rap1 activation downstream of Tie2 in ECs.

Overall, this study revealed that Rap1 is an essential molecular intermediate for the stabilizing effects of Ang-1 at cell–cell junctions of ECs. These effects of Ang-1 mediated by Rap1 at intercellular junctions have consequences on EC sprouting. This study also revealed that Ang-1 does not utilize Rap1 to mediate its adhesive effects at the cell–matrix interface of ECs. Thus, Rap1 is central for the proangiogenic effects of Ang-1 on ECs.

## Figures and Tables

**Figure 1 cells-09-00155-f001:**
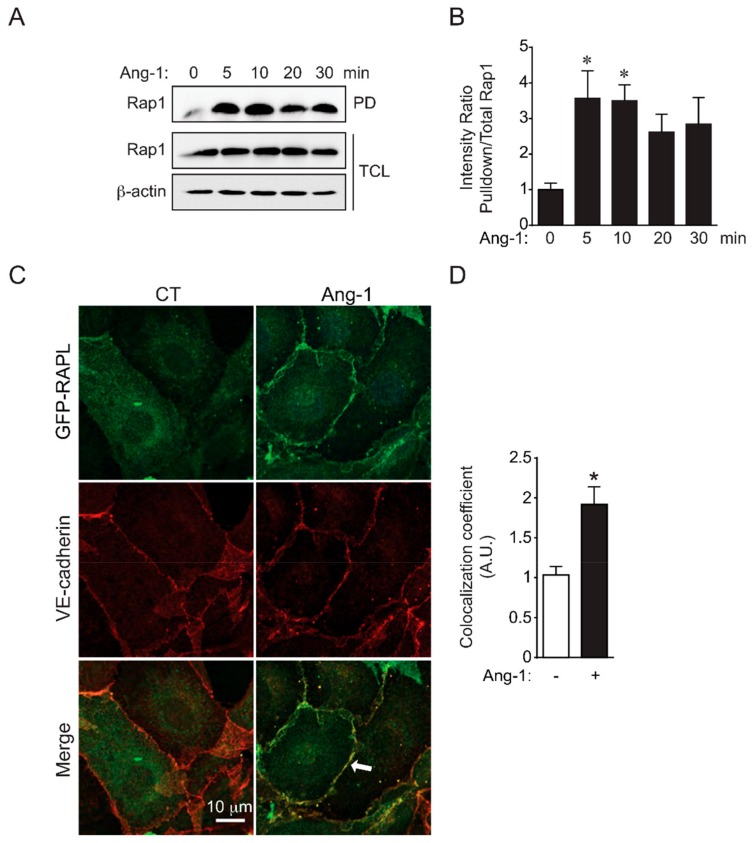
Ang-1 activates Rap1 and induces RAPL accumulation at cell–cell junctions. (**A**) Rap1 activation determined by pull-down assays using RalGDS Rap-binding domain (RalGDS-RBD/GST). Bovine aortic endothelial cells (BAECs) were treated with Ang-1 (100 ng/mL) for the indicated times, and pull-down (PD) assay were performed on protein extracts to reveal the amount of active Rap1 (Rap1-GTP) in comparison to the Rap1 amount in total cell lysates (TCL). β-actin was used as loading control. Immunoblots are representative of three independent experiments. (**B**) Densitometric quantification of Rap1 activation shown in (**A**). Average ratio of active Rap1 (PD) to Rap1 levels in TCL for three experiments; * *p* < 0.05 vs. unstimulated (0 min). (**C**) Ang-1 BAEC stimulation induces translocation of the Rap1 effector RAPL to vascular endothelial-cadherin (VE-cadherin) positive cell–cell junctions. Representative confocal micrograph of BAECs transfected with GFP-RAPL (green), treated or not with Ang-1 (100 ng/mL) for 30 min, and processed for VE-cadherin immunofluorescence (red). Recruitment of GFP-RAPL to cell–cell junctions visualized by its colocalization (merge: yellow) with VE-cadherin staining. Arrow: colocalization at cell–cell junctions. (**D**) Quantification of colocalization coefficient of GFP-RAPL and VE-cadherin expressed as arbitrary units (AU). Average colocalization coefficient of at least 30 cell junctions per condition was quantified in three different experiments; * *p* < 0.05.

**Figure 2 cells-09-00155-f002:**
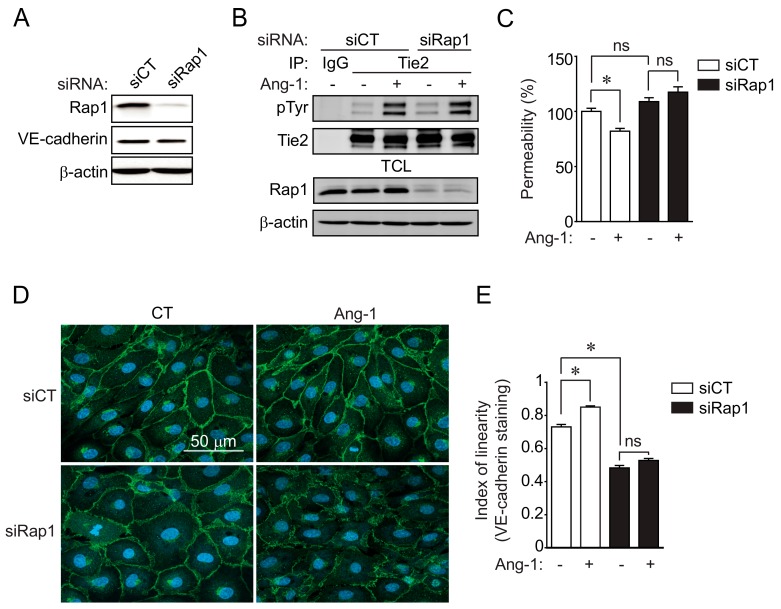
Rap1 is essential for Ang-1-induced cell-junction stabilization and endothelial-permeability inhibition. (**A**) siRNA-mediated downregulation of Rap1 in BAECs. BAECs were transfected with control siRNA (siCT) or targeting Rap1 (siRap1), and downregulation was confirmed by immunoblotting against Rap1. VE-cadherin and β-actin were used as loading controls. (**B**) BAECs transfected with siCT or siRap1 were stimulated with Ang-1 (100 ng/mL) for 30 min. Tie2 tyrosine phosphorylation levels were monitored in Tie2 immunoprecipitates (IP). Equal immunoprecipitation levels from BAEC lysates and Rap1 downregulation were confirmed by Western blot. This experiment was repeated three times with identical results. (**C**) Transendothelial permeability was determined by measuring the passage of fluorescein isothiocyanate (FITC)-dextran through BAEC monolayer. Passage of FITC-dextran was measured after exposure of BAECs, transfected with siCT or siRap1, to Ang-1 (100 ng/mL; 30 min). Graph presents average of three independent experiments (mean +/– SEM; *p* < 0.05). (**D**) Representative confocal micrographs of BAECs stained for VE-cadherin (green) and with DAPI (nucleus, blue), and transfected with siCT or siRap1. Where indicated, cells were stimulated with Ang-1 (100 ng/mL) for 30 min prior to fixation. Linearity of cell–cell junction staining (VE-cadherin) was indicative of junction stability. (**E**) Quantification of “index of linearity” as described in Material and Methods. Graph represents average index of linearity +/– SEM of at least n = 30 junctional sections between cells per conditions; * *p* < 0.05); ns: not significant.

**Figure 3 cells-09-00155-f003:**
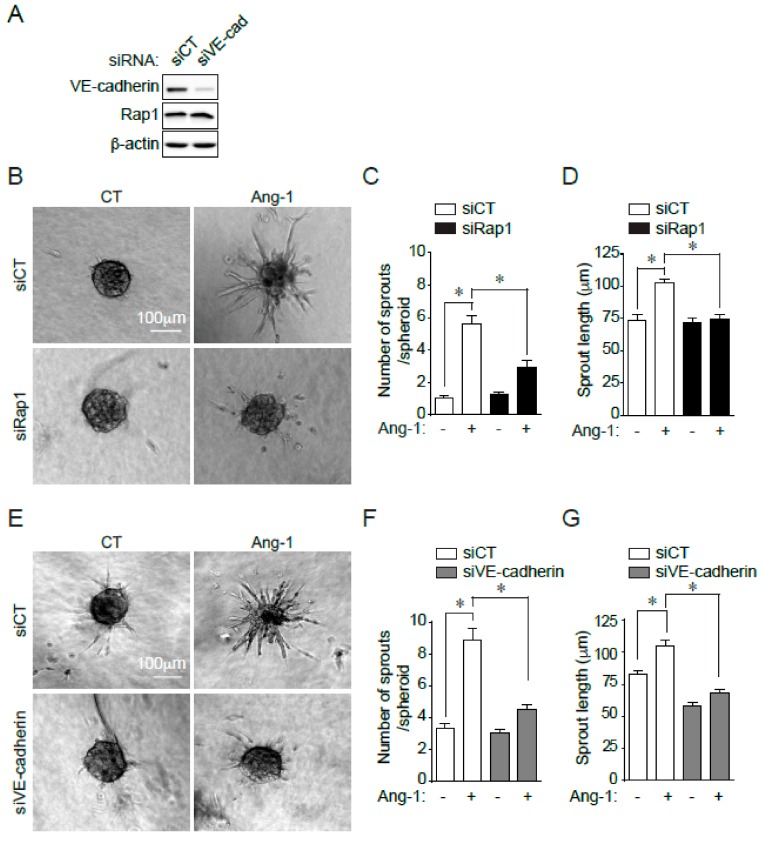
Rap1 and VE-cadherin required for Ang-1-induced BAEC sprouting. (**A**) SiRNA-mediated downregulation of VE-cadherin in BAECs. BAECs were transfected with control siRNA (siCT) or targeting VE-cadherin (siVE-cadherin), and downregulation was confirmed by immunoblotting against VE-cadherin. Rap1 and β-actin were used as loading controls. (**B**) Representative images of phase-contrast micrographs of BAEC spheroids transfected with control siCT or targeting Rap1 (siRap1) and treated with Ang-1 (100 ng/mL) or left untreated (CT). (**C**,**D**) Quantification of number of sprouts per spheroid and sprout length. Data presented as mean ± SEM of three experiments with at least eight spheroids per condition. Scale bar represents 100 μm. (**E**) Representative images of phase-contrast micrographs of BAEC spheroids transfected with siCT or siVE-cadherin and treated with Ang-1 (100 ng/mL) or left untreated (CT). (**F**,**G**) Quantification of number of sprouts per spheroid and sprout length. Data represented as mean ± SEM of three experiments with at least eight spheroids per condition. Scale bar represents 100 μm; * *p* < 0.05.

**Figure 4 cells-09-00155-f004:**
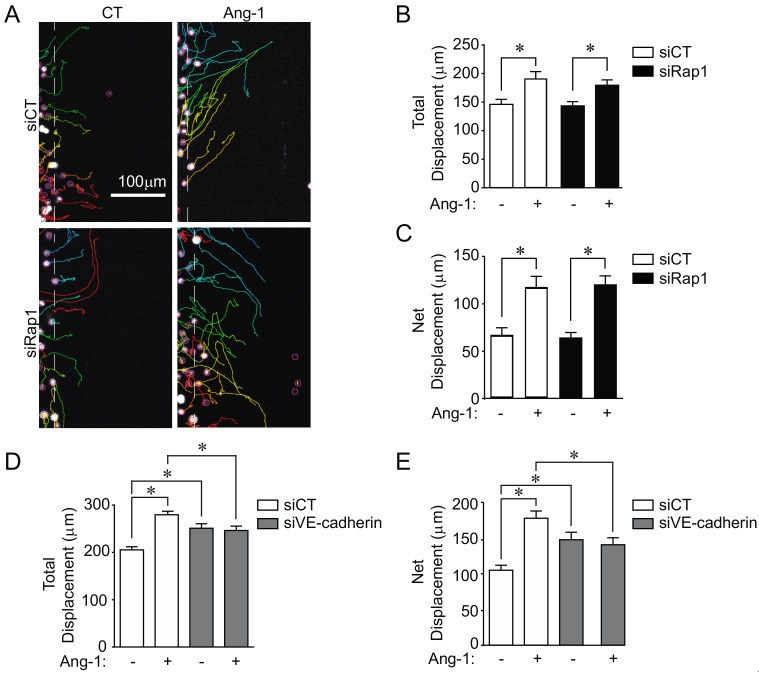
VE-cadherin, but not Rap1, is essential for Ang-1-induced BAEC migration. (**A**) Representative images of trajectories of migrating BAECs captured by live-cell imaging during 6 h of migration. BAECs were transfected with control siRNA (siCT) or targeting Rap1 (siRap1) and treated with Ang-1 (100 ng/mL), where indicated. (**B**,**C**) Quantifications of total and net cell migration. Graphs are representative of three independent experiments, each yielding similar results. At least 60 cells per experiment were quantified (* *p* < 0.05). (**D**–**E**) BAECs were transfected with control siCT or targeting VE-cadherin (siVE-cadherin), and treated with Ang-1 (100 ng/mL) where indicated, showing quantifications of total and net cell migration. Graphs representative of three independent experiments, each yielding similar results. At least 60 cells per experiment were quantified; * *p* < 0.05.

**Figure 5 cells-09-00155-f005:**
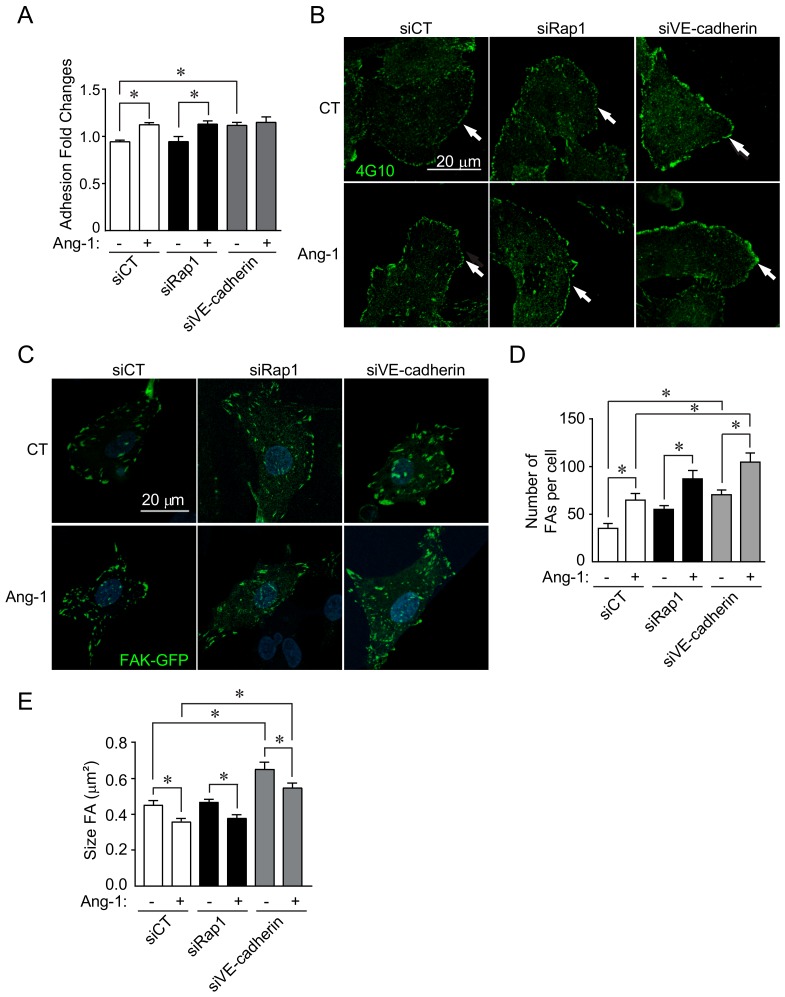
VE-cadherin required for Ang-1-induced cell adhesion to extracellular matrix. (**A**) Quantification of relative cell adhesion to gelatin-coated substrates of BAECs transfected with siRNA against VE-cadherin (siVE-cadherin), Rap1 (siRap1), or control siRNA (siCT), as indicated. Adhesion assays were performed in absence or presence of Ang-1 (100 ng/mL). Results shown as relative ratio to untreated and siCT-transfected cells. (**B**) Representative confocal micrographs of BAECs stained for total phosphotyrosine (4G10; green) and transfected with siCT, siVE-cadherin, or siRap1. Where indicated, cells were stimulated with Ang-1 (100 ng/mL) for 30 min prior to fixation. Arrows point towards cell membrane where phosphotyrosine staining was strongest. Scale bar: 20 μm. (**C**) Representative confocal micrographs images of BAECs transfected with GFP-FAK plasmids and transfected with siCT, siVE-cadherin, or siRap1. Where indicated, cells were stimulated with Ang-1 (100 ng/mL) for 30 min prior to fixation. (**D**,**E**) Quantification of number of focal adhesions (FAs) per cell and FA size in cells shown in **C**. Total of five cells per condition were counted. Graphs are representative of three independent experiments each yielding similar results; * *p* < 0.05.

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
