# Peer review of "Rap1 Is Involved in Angiopoietin-1-Induced Cell-Cell Junction Stabilization and Endothelial Cell Sprouting"

_cells, 2020, doi:10.3390/cells9010155_

Round 1

Reviewer 1 Report

The manuscripts shows nicely that Angiopoietin-1 (Ang1) activates the G protein Rap1, which is a well known regulator of endothelial cell biology. The protective effect of Ang1 reducing endothelial permeability was partially abolished by silencing Rap1. Also the sprouting angiogenesis induction by Ang1 was inhibited by silencing Rap1 but not migration.

Major points:

A major weakness of this study is the use of just a single siRNA to silence Rap1 or VE-Cadherin. At least two independent siRNA molecules should be used or alternatively Crispr/Cas approaches. Otherwise the observed effects could in the worst case just be some off-target effects.

The second major weakness of this manuscript is the lack of further mechanistic insights. It remains completely unknown via which receptor Ang1 acts on Rap1 (Tie1, Tie2, integrins?). It might by out of scope to also analyze the GEF acting between the receptor and Rap1 but at least a thorough analysis of the Ang1 receptor mediated effects an Rap1 are mandatory.

Does silencing of Rap1 increase permeability (Fig. 2B)?

Angiogenesis does not only depend on migration and cell junction remodeling but also on proliferation. How do Ang1, Rap1 and VE-Cadherin affect endothelial cell proliferation?

Author Response

Reviewer 1

The manuscript shows nicely that Angiopoietin-1 (Ang1) activates the G protein Rap1, which is a well known regulator of endothelial cell biology. The protective effect of Ang1 reducing endothelial permeability was partially abolished by silencing Rap1. Also the sprouting angiogenesis induction by Ang1 was inhibited by silencing Rap1 but not migration.

We thank the reviewer for his insightful comments.

A major weakness of this study is the use of just a single siRNA to silence Rap1 or VE-Cadherin. At least two independent siRNA molecules should be used or alternatively Crispr/Cas approaches. Otherwise the observed effects could in the worst case just be some off-target effects.

We have validated all our data with the use of two independent siRNA for Rap1 but had decided to show only one of them for clarity. We have now added the sequences of for both siRNA in the Methods section and a supplemental figure showing their efficacy has been added (Supplemental Figure 1). Two different siRNAs against VE-cadherin were also used; one against the bovine form, tested in bovine aortic endothelial cells (BAECs), and one against the human version tested in human umbilical vein endothelial cells (HUVECs). For the sake of clarity, we have decided not to present the results obtained in HUVECs, where similar results were obtained.

The second major weakness of this manuscript is the lack of further mechanistic insights. It remains completely unknown via which receptor Ang1 acts on Rap1 (Tie1, Tie2, integrins?). It might be out of scope to also analyze the GEF acting between the receptor and Rap1 but at least a thorough analysis of the Ang1 receptor mediated effects are Rap1 are mandatory.

Multiple studies show that inhibition of Tie2 in endothelial cells (ECs) completely abrogates the effects of Ang1 demonstrating that Tie2 is the receptor responsible for the effects of Ang1 on endothelial cells (Harel S. et al. Vascul. Pharmacol. 2017). Furthermore, we show in this study that Rap1 is not involved in Ang1-stimulated adhesion of ECs to the extracellular matrix. These result rule out integrins as mediators of Rap1 activation by Ang1 in ECs. In this revised manuscript we now provide results demonstrating that Ang1 stimulation of ECs induces autophosphorylation of Tie2 on tyrosine residues and that downregulation of Rap1 did not affect the activation of Tie2 by Ang1. These results confirm that Ang1 activated Tie2 in ECs and that Rap1 acts downstream of the Ang1 receptor. These results are now presented in the new Figure 2B.

We agree that it would be interesting to identify the GEF acting downstream of Tie2; we did mention this point in our discussion and proposed a few potential candidates (lines 409 – 416). We also believe that this would merit further investigations that are beyond the scope of the present study.

Does silencing of Rap1 increase permeability (Fig. 2B)?

Transfection of siRap1 in ECs does not increase significantly permeability compared to cells transfected with siCT. This information has now been added to the new Figure 2C.

Angiogenesis does not only depend on migration and cell junction remodeling but also on proliferation. How do Ang1, Rap1 and VE-Cadherin affect endothelial cell proliferation?

We agree that proliferation of ECs is an essential component of angiogenesis. However, Ang1 is well-recognized as a very weak mitogen on ECs compared to other angiogenic factors such as VEGF and bFGF. This is the main reason why we did not examine the role of Rap1 on Ang1 stimulated proliferation of ECs. In addition, VE-cadherins are involved in contact-mediated inhibition of cell proliferation, which may induce confounding results and complicate their interpretation. Thus, we decided to limit our study the role of Rap1 in Ang1-induced cell junction remodeling, migration and sprouting of ECs.

Reviewer 2 Report

In the present study the authors investigate the contribution of Rap1 to Arg-1-stimulated angiogenic effects on endothelial cells (ECs). This is an interesting study that may shed some light on the Rap1 role in the intercellular junctions and angiogenic EC properties. Nevertheless, I have some concerns on the findings.

Please, find my comments below.

Matherial and Methods

Line 68 page 2 – please correct 100 …..g/mL 

Line 95 page 3 – please correct anti--actin (Cell Signaling). 

In the Western Blot analysis please provide more details about the cells treatment and protein extraction

2.5 Immunofluorescence Microscopy - how many cells were cultured onto coverslips?

Line 117 Escherichia colishould be italic

Brands should be described as (Brand, City and Country) in all the methodology section

Line 139 – CO2 should be correct to CO2

Line 144 – Which statistic method was applied? Please describe

In each experiment please indicate how many experiments were performed (n)

Line 157 – please correct to 50,000 cells 

Line 164 – GraphPad Prism 5 software (brand, city, country)

Results:

Figure 1 – I cannot see the western blot bands (maybe a pdf format problem)

Figure 2 - Is it possible to add a cell permeability image bellow the bars in image 2B

Please standardize por P?

Figure 3A - I cannot see (maybe a pdf problem???)

Figure 3 B – please insert the size bars in the images

Figure 3 E – I cannot see (maybe a pdf problem???)

Why the authors chased BAEC as an endothelial cell model?

Page 12 Discussion – please include the study limitations

Author Response

In the present study the authors investigate the contribution of Rap1 to Arg-1-stimulated angiogenic effects on endothelial cells (ECs). This is an interesting study that may shed some light on the Rap1 role in the intercellular junctions and angiogenic EC properties. Nevertheless, I have some concerns on the findings.

Thank you for your thorough reading and analysis of our study. Unfortunately, it seems that there were many problems with the PDF version of your copy of the manuscript. We have reverified all our figures to confirm that they were complete and all images visible. Also, the PDF copies of the other reviewers seemed to be complete. Please refer to the journal’s editorial office if some of the figures are still problematic.

Please, find my comments below.

Material and Methods

Line 68 page 2 – please correct 100 …..g/mL 

Corrected.

Line 95 page 3 – please correct anti--actin (Cell Signaling). 

Corrected.

In the Western Blot analysis please provide more details about the cells treatment and protein extraction.

Methods for Western blot analyses are now described.

2.5 Immunofluorescence Microscopy - how many cells were cultured onto coverslips?

100,000 cells were initially seeded on the coverslips. This is now indicated.

Line 117 Escherichia coli should be italic

Corrected.

Brands should be described as (Brand, City and Country) in all the methodology section.

This is now done throughout the Material and Methods section.

Line 139 – CO2 should be correct to CO2

Corrected

Line 144 – Which statistic method was applied? Please describe

Statistical analyses were done as described in section 2.10.

In each experiment please indicate how many experiments were performed (n)

“n” for each experiment are all indicated in the Figure legends

Line 157 – please correct to 50,000 cells 

Corrected.

Line 164 – GraphPad Prism 5 software (brand, city, country)

This is now indicated (GraphPad, San Diego, USA).

Results:

Figure 1 – I cannot see the western blot bands (maybe a pdf format problem)

Bands are present, PDF problem.

Figure 2 - Is it possible to add a cell permeability image bellow the bars in image 2B

The confocal microscopy images showing VE-cadherin staining in Figure 2C (now 2D) reflect the permeability results.

Please standardize p or P?

Standardized to “p

Figure 3A - I cannot see (maybe a pdf problem???)

The western blots are present, PDF problem.

Figure 3 B – please insert the size bars in the images

Size bars are present, PDF problem.

Figure 3 E – I cannot see (maybe a pdf problem???)

Micrographs are present, PDF problem.

Why the authors chose BAEC as an endothelial cell model?

BAECs are primary cultures of endothelial cells that are easily amenable to transfection of siRNA, which allows for a significant downregulation of the protein of interests. Also, primary BAECs respond well to growth factors, such as Ang-1 and VEGF, and can be used for multiple types of angiogenesis tests in vitro. They strongly express endothelial cell markers, such as VE-cadherin, PECAM and eNOS, even after multiple passages in cultures. We strongly prefer using primary BAECs to the use of transformed human endothelial cells that may have lost many “endothelial properties”. Most of the results presented herein have also been validated in human umbilical vein endothelial cells (HUVECs).

Page 12 Discussion – please include the study limitations

The main limitation of this study is that experiments were done in vitro. The results should be validated in mice using Rap1 deficient mice (Ref. 29). This point is now mentioned in the discussion.

Reviewer 3 Report

This is a well written scientific manuscript. Although I am not a basic scientist I do have few general questions about the topic:

1)was there a particular reason that bovine aortic endothelia cells was analyzed versus EC from other arterial system?

2)Does the finding of Rap1 and Ang-1 in this study applicable to non-endothelial cells?

Author Response

This is a well written scientific manuscript. Although I am not a basic scientist I do have few general questions about the topic:

1)was there a particular reason that bovine aortic endothelia cells were analyzed versus EC from other arterial system?

As mentioned in the response to Reviewer 2, BAECs are primary cultures of endothelial cells that are easily amenable to transfection of siRNA, which allows for a significant downregulation of the protein of interests. Also, primary BAECs respond well to growth factors, such as Ang-1 and VEGF, and can be used for multiple types of angiogenesis tests in vitro. They strongly express endothelial cell markers, such as VE-cadherin, PECAM and eNOS, even after multiple passages in cultures. We strongly prefer using primary BAECs to the use of transformed human endothelial cells that may have lost many “endothelial properties”. Most of the results presented herein have also been validated in HUVECs.

Control of vascular permeability is mostly done in the microvasculature, not in large conduit arterial vessels. We agree that the use of endothelial cells from microvascular origin rather than from the aorta to test the anti-permeability effects of Ang-1 would be more physiologically representative. However, our long-standing experience with the use of endothelial cells in culture allows us to conclude that intracellular signaling mechanism in ECs from macro- or micro-vessels are relatively similar when studied in vitro.

2)Does the finding of Rap1 and Ang-1 in this study applicable to non-endothelial cells?

Rap1 has been shown to be an essential modulator of cell-matrix and cell-cell adhesion in many cell types and in different species, including epithelial and endothelial cells (See Reference 7). This is now mentioned in the introduction (line 44).

However, the effects of Ang-1 are mostly specific for endothelial cells since the Tie2, the Ang-1 receptor, is predominantly expressed on endothelial cells and in a minor subset of monocytic cells. Thus, this study reveals that Rap1 controls the important cell adhesive actions of another angiogenic factor, Ang-1, in endothelial cells.

Round 2

Reviewer 1 Report

All of my concerns have been addressed. There are no further issues.